# Groundwater Quality Modeling and Mitigation from Wastewater Used in Irrigation, a Case Study of the Nile Delta Aquifer in Egypt

**DOI:** 10.3390/ijerph192214929

**Published:** 2022-11-13

**Authors:** Isamil Abd-Elaty, Shaimaa M. Abd-Elmoneem, Gamal M. Abdelaal, Jakub Vrána, Zuzana Vranayová, Hany F. Abd-Elhamid

**Affiliations:** 1Water and Water Structures Engineering Department, Faculty of Engineering, Zagazig University, Zagazig 44519, Egypt; 2Belbis Engineering High Institute, Belbis 44519, Egypt; 3Institute of Building Services, Faculty of Civil Engineering, Brno University of Technology, 602 00 Brno, Czech Republic; 4Institute of Building Construction, Faculty of Civil Engineering, Technical University of Košice, 04200 Kosice, Slovakia; 5Department of Environmental Engineering, Faculty of Civil Engineering, Technical University of Košice, 04200 Kosice, Slovakia

**Keywords:** wastewater, drains, irrigation, groundwater quality, farmers’ health and life

## Abstract

Groundwater is an essential freshwater source because traditional sources of freshwater, such as rainfall and rivers, are unable to provide all residential, industrial, and agricultural demands. Groundwater is replenished by different sources: rivers, canals, drains, and precipitation. This research aims to apply numerical models for a real case study (Bahr El Baqar drain) in the Eastern Nile aquifer to monitor groundwater quality due to the use of wastewater from drains directly in irrigation due to the shortage of freshwater in this area. In addition, the effect of over-pumping from the aquifer is studied to show the extent of contaminants in groundwater. Moreover, a management strategy was achieved through mixing treated wastewater with freshwater to reduce the contamination of groundwater and overcome water shortage. Visual MODFLOW is used to simulate groundwater flow and contaminant transport into the Eastern Nile aquifer (ENDA), Egypt. In this study, three stages including 15 scenarios (five scenarios for each stage) were settled to achieve the study objectives. The first stage was carried out to investigate the impact of using untreated wastewater for irrigation due to the shortage of freshwater in this area. The results of this stage showed that increasing the use of untreated wastewater increased the contamination of the aquifer. The average COD concentrations in the five scenarios reached 23.73, 33.76, 36.49, 45.13, and 53.15 mg/L. The second stage was developed to evaluate the impact of over-pumping and using untreated wastewater for irrigation due population increase and a reduction of freshwater in the Nile Delta. The results revealed that over-pumping has increased the contamination of the aquifer and the average COD concentrations increased to 25.3, 33.34, 40.66, 48.6, and 54.17 mg/L. The third stage was applied to investigate the impact of mixing treated wastewater with freshwater for irrigation to support the freshwater quantity. The results of this stage led to enhanced water quality in the aquifer and the average COD concentrations decreased to 20.26, 23.13, 26.03, 30, and 32.83 mg/L. The results showed that mixing freshwater with treated wastewater has a good influence on water quality, can be safely used in irrigation and reduces the effects on farmers’ health and life.

## 1. Introduction

The global water distribution has a number of forms, about 97.5% of the earth’s water being saltwater in seas and oceans and only 2.5% being freshwater. Most of this 2.5% of freshwater is confined and locked up as glaciers and ice caps 68.7% and deep groundwater 30.1%, with only 1.2% for surface freshwater. Surface freshwater sources, such as rivers and lakes, account for just around 93,100 cubic km of freshwater. These percentages demonstrate how little freshwater mankind possesses and the need for conserving it [1]. The available amount of freshwater on Earth is sufficient to meet human needs. However, only around 1% of the total freshwater quantity is reachable for human use. The agricultural sector consumes more than 70% of the available freshwater. Water scarcity in developing nations is caused by industrial effluent contamination of surface and groundwater, whereas sewage from urban and built-up regions is a main source of water degradation. Industrial wastewater and sewage can both be used for these requirements as the application of water recycling overcomes the political and social strain on current water sources while reducing wastewater discharge burden to the soil [2].

The problem is that water is not distributed uniformly; by the year 2025, almost 60% of the world population may face water shortage. Countries experiencing water constraint have a tendency to create non-traditional water sources. One of the unconventional water resources is the reuse of drainage water. Food and agriculture consume up to 70% of Egypt’s water supplies, making them the country’s major consumers of water. Although the drainage water is rather good quality, around 25% of the irrigation water flows to the drains. The official reuse of drainage water has grown from 2.6 billion m^3^ in 1988/1989 to 5.0 billion m^3^ in 1998/1999, with a large quantity (between 2.8 and 4.0 billion m^3^) of the unofficial drainage water being utilized by farms [3]. The wastewater contributes with partial assistance to compensate water shortage. Nearly 40% of the world’s population suffers from lack of water needed for irrigation, which causes agricultural losses. As a result of the water shortage, especially in arid and semi-arid regions, wastewater is used as the main water source, and this is due to its large quantity, as wastewater discharge reaches 400 BCM/yr globally. The concentration of wastewater varies from raw to dilute due to the different disposal outlets: for example, urban wastewater from domestic, commercial, and industrial drainage; the surface runoff of rainwater; treated wastewater through treatment plants and finally reclaimed or recycled water. Sewage water is considered a stable source of water because it is not dependent on climatic conditions or rain, as it is easy to use for the agriculture throughout the year. In addition, it contains nutrients that could help the crops to grow and in turn reduces fertilizer costs and the use of chemicals [4].

Wastewater, in general, refers to liquid waste released from residential, commercial and industrial organizations, which may also be combined with storm water if it is present [5]. Over decades, wastewater has been used as a supplier of agricultural nutrients in developing nations. Before the development of wastewater treatment methods to avoid water contamination, wastewater was discharged into agricultural fields in many European and North American regions. Although wastewater usage in agriculture offers significant benefits, it may also cause significant threats to human health, particularly when untreated wastewater is utilized for agricultural irrigation. Microbial and chemical threats to the public health are the most serious. Wastewater consumption in agriculture can potentially pose many environmental problems by contaminating soil and groundwater. Farmers are frequently forced to utilize untreated wastewater because wastewater treatment is unavailable, and freshwater is either inadequate or prohibitively costly. However, using wastewater for irrigation could have a number of environmental, agricultural, and water resource management positive effects if it is properly designed, executed and managed [6]. About 200 million villagers globally irrigate at least 20 million hectares (ha) with raw or partly treated wastewater [7], representing 8% of the total global irrigated farmland. Irrigating urban fields using untreated or partly treated wastewater has a number of advantages and disadvantages. The benefits, especially in poor nations, include water saving [8], recycling of nutrients, reducing fertilizer need, and wastewater treatment on land, irrigation water availability [7], reducing the need for expensive transport or storage equipment’s, increasing nutrition for urban residents and improve living. The negatives include higher health risks and lower environmental quality as water, soil, and crops become progressively polluted with pathogens, metals, and other contaminants [9].

The protection of reuse the wastewater is a critical issue for crop irrigation around the world. If wastewater is not properly treated until being used for irrigation, it can damage the soil (salinization, toxicity from sodium, boron ions, and chloride; decreased aeration and pore clogging from suspended solids in wastewater; structural deterioration; and reduced hydraulic conductivity) as well as agricultural production (excess nutrients cause heavy metal accumulation, biological load, and delayed or erratic growth of the plant), in relation to groundwater (by seeping of unnecessary nitrates). Where possible, treated wastewater should be reused, and drainage methods should mitigate the potential negative impacts on the atmosphere and public health. The most appropriate treatment system for drainage before it is used for irrigation is one that produces an effluent that satisfies quality criteria from a microbiological and chemical standpoint while requiring minimal operation and maintenance [10]. According to the human health, there are possible biological components (e.g., protozoa, viruses, helminths, and enteric bacteria) that might endanger farmers’ health if they contact directly with raw wastewater. Furthermore, heavy metals, pesticides, and pharmaceutical compounds accumulated in grown plants watered with wastewater, posing a risk to consumers [11].

Egypt relies on reusing agricultural drainage water to fill the significant water shortage. The most economically and practically feasible solution for Egypt to close the gap between available water and demands is the reuse of drainage water. Reusing agricultural drainage water increased Egypt’s water supply by 20% [12]. The scarcity of freshwater resources became a global problem in the twenty-first century and the water resources have become limited, whether the share of the Nile River or non-renewable groundwater and rainwater; at the same time, the demand for water has increased and become ongoing for various activities such as industrial, agricultural, commercial, domestic and environmental use as well as recreational [13]. The Nile aquifer system is the most significant aquifer in Egypt. Agriculture percolation and infiltration through irrigation and drainage canals are the primary sources of aquifer recharge. The pollution of aquifers is strongly associated with contaminated surface water caused by the interaction of groundwater with surface water. Pollution is worse in the Nile Valley’s edges and desert fringes as well as in the shallow sections of the aquifers [14].

A number of research studies have been carried out to assess the impacts of using drainage water for irrigation in the Nile Delta. Fifty groundwater samples and 79 surface water samples from drains were collected to investigate the quality of the Nile Delta aquifer. The well samples were gathered from the top portion of the aquifer system, at depths ranging from 30 to 135 meters below the surface. Screens in studied wells were often found in the water-bearing zones of the top half, no more than 150 m below the surface of the ground. The studies showed that iron and manganese concentrations were higher in the ancient lands because of the overall natural characteristics of the Nile Delta aquifer, and this was especially true in places with a clay cap. The extracted groundwater can be aerated to remedy this condition. Moreover, half of the samples had significant levels of aluminum [15]. In the southern section of Lake Manzala, the impact of drain effluent was investigated for changes in trace metals, inorganic anions, and cations in water. In the region receiving urban and agricultural wastewater from Bahr El Baqar drain, the study found a clear depletion of dissolved oxygen as well as a significant rise in COD and BOD. Due to the irregular inflow of various wastes, the concentrations of nutritional salts indicated a vast range of fluctuation and rapid changes. Ammonia, nitrate, and orthophosphate concentrations were also found to be excessive. In regions that received residential and agricultural wastewaters, the quantities of trace metals in lake water were significant [16,17].

Recently, some studies assessed the impact of recharging groundwater by contaminated water from irrigation. They used MODFLOW to simulate different scenarios of pumping rates and wastewater recharge for a hypothetical case study. The results revealed that upon increasing the pumping rates, wastewater recharge rates had a detrimental influence on groundwater quality and increased pollutants in the aquifers [18,19]. In addition, MODFLOW was used to assess the impact of different pumping rates and seepage from open contaminated drains on groundwater quality. The findings indicated that excessive pumping has a significant negative impact on groundwater contamination [20]. The benefits of using wastewater for irrigation include using nutrients found in wastewater and replenishing groundwater reservoirs to compensate for the lack of freshwater. As a result, less fertilizer is used, and less groundwater needs to be treated. Due of this, aquatic bodies receive less fertilizer input and have less direct outflow [21]. In the Nile delta, a numerical study is conducted for groundwater protection from the seepage of open drains using a cutoff wall and lining. The results reveal that lining the drains could protect groundwater from pollution and installing cutoff has a positive impact on shallow aquifers but has no impact on deep aquifers [22].

In this study, the numerical model MODFLOW is used to study a real-world case study of the Eastern Nile delta aquifer (ENDA) in Egypt to monitor the impact of using wastewater from Bahr El Baqar drain in irrigation on the groundwater quality on the ENDA. Three stages were applied to achieve the specific objectives of the study, including the impact of using untreated wastewater from the drain for irrigation on groundwater quality, impact of increasing abstraction rates with using untreated wastewater on groundwater quality and the impact of using treated wastewater from the drain for irrigation on groundwater quality. The scenario of using treated wastewater mixed with freshwater for irrigation is examined to reduce the groundwater contamination and protect farmers’ health and life.

## 2. Materials and Methods

### 2.1. Study Area

The study area of the eastern Nile Delta is located between latitudes 30°00′ and 31°30′ N and longitudes 31°00′ and 32°30′ E and covers an area of approximately 15,000 square kilometers (km^2^) [23]. As shown in Figure 1a [24], the study area is boarded by the Mediterranean Sea and Lake Manzala from the north, the Suez Canal from the east, Cairo Suez Desert Road from the south, and the Damietta branch from the west [23]. The elevation of the study area ranges from 1 to 50 m above the mean sea level [25]. The climate of the study area is desert climate, with cold winters and sunny summers [12]. In the coastal regions, the annual temperature varies from 37 to 14 °C. Temperatures in the deserts varied from 46 °C during the day to 6 °C at the night [26,27]. The intensity of rainfall in the study area is very limited because it is located in an arid climatic condition. The minimum average annual precipitation in Cairo is 25 mm/yr, and the maximum average annual precipitation in coastal area increase to 133 mm/yr. The precipitation rises from south to north, with only a limited impact on groundwater recharge, apart from the Ismailia region [28]. The rate of evaporation in Cairo is 142 mm/yr, and the rate of evaporation is decreased in the north and east to be 44 mm/yr at Mansoura. As a result of high wind speed and temperature, the evapotranspiration has increased in southward in Suez, and high values of evapotranspiration at Port Said [28]. These factors have affected the water resources in the Nile Delta, and farmers started using wastewater for irrigation to overcome the water shortage. 

The study area’s stratigraphy is categorized into Pliocene sediments, which include early and late Pliocene. These sediments is formed from Kafr El-sheikh, Abu-Madi, and El-wastani formations. The other sediments located unsettlingly on the top of the pliocene sediments are the Quaternary or (Pleistocene and Holocene) [29]. The components of the early Pleistocene sediments are the ancient Delta sediments (flinty pebbles, loose coarse-grained quartz sands, gravel). The Pleistocene deposits serve as a major aquifer unit in the study area from the hydrogeological standpoint. The Pliocene plastic clay layer lays underneath the aquifer, where it behaves as an aquitard in the flood plain area, where it exists as an insulator of the aquifer layer due to impact of old formations. The Tertiary aquifers of Miocene and the Oligocene ages are covered by the Quaternary aquifer, especially in the east and south regions [30] (see Figure 1b) [13]. The formation of the top boundary of the deltaic deposits originates from the Holocene and is composed of silt and clay that is semi-permeable. The average hydraulic conductivity for the clay cap ranges between 50 and 500 mm/day in the horizontal direction and 2.5 mm/day in the vertical direction. The porosity of the aquifer varies between 25 and 40%. In the south and the middle of the Delta, the clay layer’s thickness ranges from 5 to 20 m, and in the north, it reaches 50 m [31]. The annual abstraction rate from the Quaternary aquifer was estimated by 4.6 × 10^9^ m^3^/yr in 2010. In Figure 2a displays the annual abstraction in (Bm^3^/yr) from 1981 to 2016. The Nile Delta aquifer is recharged by leakage from excess irrigation and leakage from the Nile River, irrigation channels and drains. The values of leakage toward the aquifer in the middle parts ranges between 0.25 and in the desert parts is 0.8 mm/day (see Figure 2b) [13]. 

### 2.2. Numerical Model

Visual MODFLOW software is used to simulate groundwater flow and solute transport in the ENDA. The partial differential equation of groundwater flow employed in MODFLOW is expressed in the form [33]:(1)$$\frac{\partial }{\partial t}\left(K_{xx}\left.\frac{\partial h}{\partial x}\right)\right.+\frac{\partial }{\partial y}\left(K_{yy}\left.\frac{\partial h}{\partial y}\right)\right.+\frac{\partial }{\partial z}\left(K_{zz}\left.\frac{\partial h}{\partial z}\right)\right.+W=S_{s}\frac{\partial h}{\partial t}$$

The transport of solute mass in groundwater is described by the following partial differential equation [34]:(2)$$\frac{\partial c}{\partial t}=\frac{\partial }{\partial x_{i}}(D_{ij}\frac{\partial c}{\partial x_{j}})-\frac{\partial }{\partial x_{i}}(V_{i}C)+\frac{q_{S}}{\theta }C_{S}+\sum_{K-1}^{N}R_{K}$$
where *K_xx_, K_yy_,* and *K_zz_:* Hydraulic conductivity along the x, y, and z coordinate axes (T^−1^), *h*: Potentiometric head (L), *W*: Volumetric flux per unit volume representing sources and/or sink of water, with *W* < 0.0 for flow out of the groundwater system, and *W* > 0.0 for flow in (T^−1^), *S_S_*: Specific storage of the porous material (L^−1^), *t*: Time (T), *C*: Groundwater concentration (mL^−3^), *D_ij_*: Dispersion coefficient (L^2^T^−1^), *V_i_*: Seepage or linear pore water velocity (LT^−1^), *q_s_*: Water flux of sources (positive) and sinks (negative) (T^−1^), *C_s_*: Sources or sinks concentration (mL^−3^), *Ɵ*: media porosity (dimensionless) and *R_k_*: Chemical reaction term (mL^−3^T^−1^). Explanation of partial differential equation and parameters of MODFLOW were presented by McDonald and Harbaugh (1988) [33].

#### 2.2.1. Model Geometry

In this study, visual MODFLOW is used to simulate the groundwater flow and solute transport in the ENDA. The total area is about 14,000 km^2^. This aquifer is bounded by the Damietta branch from the west, the Mediterranean Sea from the north, the Suez Canal from the east and Ismailia canal from the southeast. The most crucial step is to create the model domain, which should encompass the whole region, including groundwater flow and pollutant transfer. The domain was split into 129 columns and 163 rows, with a cell area of 1.00 km^2^. Figure 3a shows the studied domain of the ENDA.

The aquifer is divided into eleven layers; the top layer is the clay cap with variable thickness starts from 20 m in the south and progressively increases to 50 m in the north. The soil type of the aquifer was represented by dividing the layers from two to eleven into equal thicknesses. Fine sand was allotted to layers from two through five. Layers (six to nine) were dedicated to coarse sand, whereas layers ten to eleven were designated to graded sand and gravel. Figure 3b and c show the vertical and horizontal sections in the ENDA.

#### 2.2.2. Boundary Conditions

There are several forms of head and contaminant boundary conditions in the study region including surface irrigation and drainage systems, rivers, lakes, the Mediterranean Sea, and the Suez Canal. These boundaries have been determined using four main water bodies including the Damietta branch in the west, El Manzala Lake and the Mediterranean Sea in the north, Suez Canal in the east and Ismailia Canal in the south as well as Bahr El Baqar, Bahr Hadous, and El Serw drains as boundary conditions. Figure 4a presents the boundary conditions of the head for the study area and location of observation wells to evaluate the COD concentration. The study focuses on the Eastern Nile delta, which has the largest and most contaminated drain (Bahr El Baqar) to evaluate solute transfer from this drain to the aquifer. The length of Bahr El Baqar drain is about 85 km with an average depth of 5 m. It is located in the center of the END region [35]. The contaminant boundary conditions were assigned by constant concentration for pollution source. COD is considered as industrial pollution with concentration equal to 112.50 mg/L along the drain length [19]. Figure 4b shows three observation wells chosen to measure the concentration of COD in aquifer distributed along the drain (start, middle, end).

#### 2.2.3. Hydraulic Parameters

The hydraulic characteristics, including hydraulic conductivity, transmissivity, specific yield, specific storage, and effective porosity, are presented to the model domain based on the previous studies by [36]. The hydraulic parameters of the aquifer include the vertical and horizontal conductivities (K_h_, K_v_) (m/day), the specific storage (S_S_) (1/m), the specific yield (S_y_) and the effective porosity (n_eff_). These parameters were provided based on previous studies as shown in Table 1 [37].

#### 2.2.4. Recharge and Abstraction

The recharge values depend on location and land use. According to [38], the recharge to or discharge from the aquifer has a significant impact on groundwater flow. The recharge is accomplished by three mechanisms in the ENDA. The first is due to seepage from canals in addition to extra water return from irrigation. The measured recharge rates in the ENDA range between 0.25 and 0.80 mm/d. The second is the rainfall, which has an annual average of 25 mm. The third is inter-aquifer flow to groundwater. The recharge values used in the model are shown in (Appendix A, Figure A1). The total recharge in the ENDA equals to 4.29 Mm^3^/d. The ENDA has a massive number of pumping wells that are scattered over the area to cover the demands for irrigation, people, and industry. The locations of wells and abstraction rates were modeled according to [15], and the overall abstraction is 3.78 Mm^3^/d. (Appendix A, Figure A2). shows a map of pumping wells distribution in the ENDA.

#### 2.2.5. Model Hydrological Properties

The main characteristics influencing the groundwater system is the conductance of canals and drains. The Eastern Nile Delta has an extensive irrigation and drainage systems, which affects flow and solute transport in the aquifer. Seepage from the canals and drainage systems is determined by the conductance of the canal, which can be estimated using the following equation.
(3)$$C=\frac{L\times W\times K}{M}$$
where *C:* is the bed conductance for the canal or drain (L^2^/T), *L*: is the canal or drain length (L), *W*: is the width of the canal bed (L), *K*: is the vertical hydraulic conductivity for the canal bed (L/T), and *M*: is the drain or canal bed thickness (L).

#### 2.2.6. Model Calibration

The calibration for groundwater head and solute transport was performed at this step. The observed head was compared with the calculated one and (COD) concentration from the model. For the piezometric head, field data were measured in 21 observation wells. The distribution and position of observation wells are shown in (Appendix A, Figure A3a). A number of 17 contamination observation wells distributed in the study area were selected to measure the groundwater contamination in aquifer (see Appendix A, Figure A3b). The calculated and observed heads in the ENDA are presented in Figure 5a. It demonstrates a residual range of (0.04 to −1.274) m with root mean square (RMS) of (0.716) m and normalization root mean square (RMS) of 5.466%. The model’s calibration objective is 10% of the difference between the highest and lowest head, which is ranged from 0 to 16.00; hence, 1.60 m is estimated. The results from the model for the calculated head gave a good match with the observed ones. The model results presented a good reality for investigating groundwater levels in the future. In the ENDA, the numerical model is used to investigate groundwater flow. The results revealed that groundwater levels ranged from 16 in the south to zero in the north. In the ENDA, the direction of velocity through groundwater flow shows the direction of flow from the highest head at the south to lowest in the north, as shown in Figure 5b. The minimum and maximum velocity in the clay cap is 0.0007 and 0.00053 m/d, respectively, whereas the average velocity through the quaternary aquifer equals 0.000615 m/d.

A MT3D model was utilized to investigate the effect of contaminated open drains on the groundwater quality. The contaminant extension was found to have distributed into the aquifer. According to the study results, the contamination from Bahr El-Baqar drain and from agriculture drain into the aquifer flowed in the same direction as the groundwater flow, which was clearly illustrated in Figure 6a; this figure presents the contamination distribution in the study area domain. The model was calibrated using 17 observation wells; the calculated values match well with the observed values as shown in Figure 6b. The difference between calculated and observed concentration is shown in the figure.

In the calibrated groundwater model, the water balance estimation is a very essential step to assess the groundwater flow system. The model has inflow and outflow systems. The inflow includes the recharge, constant head, and the leak from the river, while the outflow includes the leakage from the drain and the pumping wells. Table 2 presented the zone budget of inflow and outflow for these elements, and the total inflow and outflow reached 4,879,520 and 4,879,588 m^3^/d respectively. The total outflow is larger than the total inflow by 68 m^3^/d, which means the discrepancy equals zero.

The solute transport models require time to reach a steady-state situation; this time is determined by the computer power and the central processing unit (CPU), model accuracy, cell diminution, and when the salt concentration remains constant over time; at this point, the model has reached the steady-state situation. The current model reached the steady-state situation after 1,825,000 days. (Appendix A, Figure A4)., shows the total salt source in and sinks out in the ENDA with the total mass of aquifer 8.77788 × 10^9^.

#### 2.2.7. Model Scenarios

Three main stages were applied to investigate the effect of using untreated wastewater for irrigation purposes including a number of 15 scenarios. In stage (1), the salt mass balance equation is used to calculate the concentration of irrigation water after mixing with fresh water. The equations can be written as follows [39]:(Mass flow rate of pollutants) in = (Mass flow rate of pollutants) out
Qw × Cw + Qus × Cus = Qds × Cds
0.9×8.833 + 0.1×112.5 = C1×1  (C1 = 19.5)
0.8×8.833 + 0.2×112.5 = C2×1  (C2 = 29.6)
0.7×8.833 + 0.3×112.5 = C3×1  (C3 = 40.0)
0.6×8.833 + 0.4×112.5 = C4×1  (C4 = 50.3)
0.5×8.833 + 0.5×112.5 = C5×1  (C5 = 60.7)

Stage (2) includes abstraction rates due to using untreated wastewater quantities, and the abstraction rates due to using treated wastewater quantities are included in stage (3). The principle of salt mass balance was used to estimate the recharge concentration after mixing treated wastewater with freshwater. The equations can be written as following:0.9×.833 + 0.1×50 = C1×1  (C1 = 13.0)
0.8×8.833 + 0.2×50 = C2×1  (C2 = 17.1)
0.7×8.833 + 0.3×50 = C3×1  (C3 = 21.2)
0.6×8.833 + 0.4×50 = C4×1  (C4 = 25.3)
0.5×8.833 + 0.5×50 = C5×1  (C5 = 29.4)
where Qw: wastewater flow rate (m^3^/d), Cw: concentration of a pollutant (mg/L), Qus: stream flow rate (m^3^/d), Cus: concentration of the pollutant (mg/L), Qds: downstream flow rate (m^3^/d), and Cds: concentration of the mix after discharge (mg/L). The three main stages and scenarios are presented in Table 3.

## 3. Results

The result of the numerical analysis of the 15 scenarios mentioned before and management of groundwater contamination in the ENDA are discussed in detail in the following subsections. Three observation wells (a, b, c) have been selected to evaluate the COD concentration along Bahr El Baqar drain in the ENDA, as shown in Figure 4b.

### 3.1. Effect of Using Untreated Wastewater for Irrigation on Groundwater Quality (Stage 1)

As a result of water shortage at the end of irrigation canals in the Nile Delta, farmers use drains water for irrigation without treatment. This stage evaluated the effect of using wastewater directly from drains without treatment for irrigation on the groundwater quality in the aquifer due to reduction in the Nile freshwater by 90, 80, 70, 60 and 50% with COD concentration of 8.83 mg/L. This shortage was covered by using untreated wastewater from drains with percentages of 10, 20, 30, 40 and 50% to substitute the reduction of freshwater and the COD concentration in drains is 112.5 mg/L. The modeled recharge concentration of COD in this stage was 19.5, 29.6, 40.0, 50.3 and 60.7 mg/L at a constant abstraction rate of 3,780,900 m^3^/d. The results indicated that increasing the use of untreated wastewater has increased the concentration of COD in the aquifer, which in turn increased the contamination of groundwater. Three observation wells (Well-a, Well-b, and Well-c) were used to evaluate the COD concentration in the ENDA (see Figure 4b). The values of COD concentration for different scenarios (2 to 6) are shown in Table 4. Figure 7 shows a comparison between the five scenarios applied in stage 1 with the base case (see Table 3). The average values of COD concentration increased to 23.73, 33.76, 36.49, 45.13 and 53.15 mg/L for the five scenarios compared with 13.1 mg/L at the base case. Figure 8 shows a vertical cross-section for COD distribution in the END aquifer for different scenarios. Substituting the shortage of freshwater of (10%) by wastewater for irrigation has doubled the concentration of COD in ground water (scenario 2); however, when there was a decrease by 50% (scenario 6), the concentration reached four times the base case (scenario 1) (see Table 4). The results reveal that COD concentration is increasing in the ENDA with increasing the use of wastewater from drains for irrigation due to the decrease of the freshwater from the Nile River, which represent a huge danger of farmers and animals’ health and life. 

### 3.2. Effect of Over-Pumping and Using Untreated Wastewater on Groundwater Quality (Stage 2)

In this stage, the impact of increasing abstraction rates is studied through increasing the rates by 20, 30, 40, 50 and 60%, while the quantities of Nile freshwater and untreated wastewater used for irrigation are the same as in stage 1, with COD concentration of 19.5, 29.6, 40.0, 50.3 and 60.7 mg/L. In addition, the three concentration observation wells (Well-a, Well-b, Well-c) were used to evaluate COD contamination in the ENDA. The COD values for different scenarios (7–11) are shown in Table 4. Figure 9 shows a comparison between different scenarios (7 to 11) of stage 2 with the base case (scenario 1). In this stage, the average values of COD concentration in the study area reached 25.30, 33.34, 40.66, 48.60 and 54.17 mg/L for scenarios (7 to 11) compared with 13.1 mg/L at the base case (scenario 1) as shown in Table 4. Figure 10 shows a vertical cross-section for COD distribution in the ENDA for different over-pumping rates and different quantities of Nile freshwater with untreated wastewater used for irrigation. Increasing abstraction from the aquifer by (20%) has doubled the concentration of COD in groundwater (25.30 mg/L) (scenario 7); however, for the decrease by 60% (scenario 11), the concentration reached (54.17 mg/L) about four times the base case (13.1 mg/L) (scenario 1) (see Table 4). The result of this stage showed that over-pumping has increased the COD concentration in the ENDA which led to an increase in the groundwater contamination. This requires the management of abstraction from the aquifer to protect the groundwater from pollution.

### 3.3. Effect of Over-Pumping Rate and Using Treated Wastewater on Groundwater Quality (Stage 3)

In this stage, the wastewater has been treated and mixed with the Nile freshwater before being used for irrigation. Because the abstraction rate is expected to increase due to the population growth, the impact of increasing abstraction rates is considered in this stage through increasing the rates by 20, 30, 40, 50 and 60%, and the treated wastewater was used for irrigation by 10, 20, 30, 40 and 50% from irrigation water, with COD concentration of 50 mg/L. Meanwhile, the Nile freshwater recharge decreased by 90, 80, 70, 60 and 50% with COD concentration of 8.83 mg/L. The mixed water of COD concentration reached 13.0, 17.1, 21.2, 25.3 and 29.4 mg/L (see Table 3). The same three observation wells (Well-a, Well-b, Well-c) have been used to evaluate the COD concentration in the ENDA. The results of scenarios (12 to 16) are shown in Table 4. The comparison between different scenarios of stage 3 (scenarios 12–16) at the three wells is presented in Figure 11. The average COD concentration in the study area reached 20.26, 23.13, 26.03, 30 and 32.83 mg/L compared with 13.1 mg/L at the base case. Figure 12 shows a vertical cross-section for COD distribution in the END aquifer for different scenarios (12 to 16) of stage 3. Increasing abstraction from the aquifer by (20%) with using treated wastewater has decreased the concentration of COD in groundwater from 25.30 mg/L (scenario 7) to 20.26 mg/L (scenario 12); however, for the decrease by 60%, the concentration decreased from 54.17 mg/L (scenario 11) to 32.83 mg/L (scenario 16), which represents a 40% improvement in water quality (see Table 4). The results indicated that the concentration of COD in the aquifer has decreased when treated wastewater is mixed with freshwater and used for irrigation in the END.

## 4. Discussion

Three observation wells a, b and c were distributed in the study area. The first well (a) was located in the lower part, the second well (b) was located in the middle part and the third well (c) was located in the upper part. The relationship between COD concentration and average values at the three wells for the base case (scenario 1) and the three stages (scenarios 2–16) are summarized in Figure 13. The results of the first stage (scenarios 2 to 6) showed an increase in the COD concentration in the ENDA for using untreated wastewater for irrigation. The results are match with [19,21] who utilized untreated wastewater in irrigation due to freshwater reductions. However, when over-pumping was added to stage one, as discussed in stage 2 (scenarios 7–11), the concentration of COD in the ENDA has increased, as shown in Figure 13. In addition, previous studies approved that over-pumping has a negative impact on groundwater quality as presented in [15,20], which matches with the results of the current study. In order to improve the quality of water and decrease the COD concentration in the ENDA, the third stage (scenarios 12 to 16) has been applied where the wastewater was treated and mixed with the Nile freshwater and used for irrigation. The scenarios of stage three from 12 to 16 showed that the COD concentration in the ENDA has decreased due to the use of treated wastewater. The results were consistent with [18,22], who revealed that mixing the untreated wastewater with freshwater or treated water led to reducing the contamination in aquifers. The results of this study recommend treating wastewater from drains and mixing with freshwater before use in irrigation. This could help protect groundwater from pollution, which protects the health and life of people and animals. 

## 5. Conclusions

Groundwater is considered an important freshwater source for many countries, but it is vulnerable to pollution from many sources including the use of untreated wastewater for irrigation, which leads to a deterioration of groundwater quality. Investigating the contamination of groundwater aquifers and protection is an important issue in the Nile Delta, Egypt. Shortage of the Nile freshwater supplies has contributed to the use of unconventional water resources including untreated wastewater for irrigation, which has affected the groundwater quality in the Nile Delta aquifer. In the current study, a numerical model is used to monitor the contamination of groundwater due to the use of wastewater for irrigation. The current study was carried out in three stages using the numerical model of visual MODFLOW for investigating the impact of using untreated wastewater for irrigation (stage 1), over-pumping with using untreated wastewater for irrigation (stage 2), and over-pumping with using treated wastewater (stage 3). The numerical model was applied to one of the extremely polluted drains (Bahr El Baqar drain) in the Eastern Nile Delta aquifer (ENDA). The results indicated that increasing the use of untreated wastewater quantities (stage 1) by 10, 20, 30, 40 and 50% with a concentration of 112.5 mg/L combined with decreasing the recharge from the Nile freshwater by 90, 80, 70, 60 and 50% with a concentration of 8.83 mg/L has led to an increase in the concentration of COD in the aquifer to 23.73, 33.76, 36.49, 45.13 and 53.13 mg/L, respectively, for scenarios (2 to 6). In stage 2, the abstraction rates increased by 20, 30, 40, 50 and 60% with the same parameters used in (stage 1). The result of average COD increased to 25.30, 33.34, 40.66, 48.60 and 54.17 mg/L, respectively, for scenarios 7 to 11. This indicates that over-pumping has led to the increase the aquifer contamination. In stage 3, the abstraction rates increased by 20, 30, 40, 50 and 60%, but treated wastewater has been used by 10, 20, 30, 40 and 50% which mixed with Nile freshwater quantity by 90, 80, 70, 60 and 50%. The average values of COD contamination decreased to 20.26, 23.13, 26.03, 30.00 and 32.83 mg/L, respectively, in scenarios 12 to 16. The results showed a positive impact of using treated wastewater that mixed with freshwater (stage 3), where the COD concentration has been decreased compared to stages 1 and 2. Treating wastewater and mixing with freshwater could reserve the groundwater storage in the aquifers and protect the health and life of people who are depending on such water.

## Figures and Tables

**Figure 1 ijerph-19-14929-f001:**
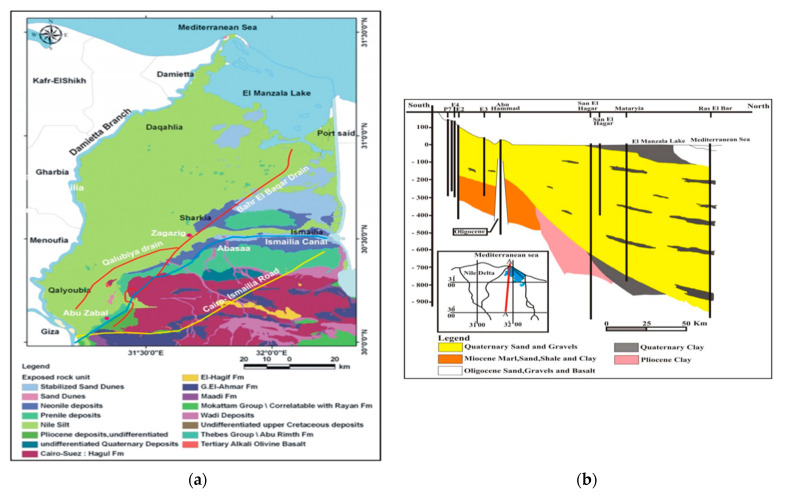
(**a**) Location map of the study area [24], (**b**) hydrogeological cross−sections of the Quaternary aquifer in the Nile Delta [13].

**Figure 2 ijerph-19-14929-f002:**
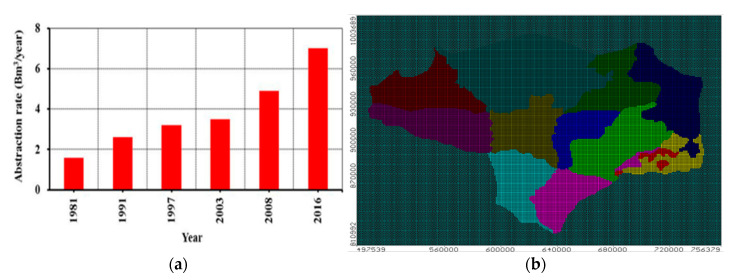
(**a**) Annual abstraction rates from the Nile Delta aquifer [32], (**b**) Recharge from the Nile Delta [13].

**Figure 3 ijerph-19-14929-f003:**
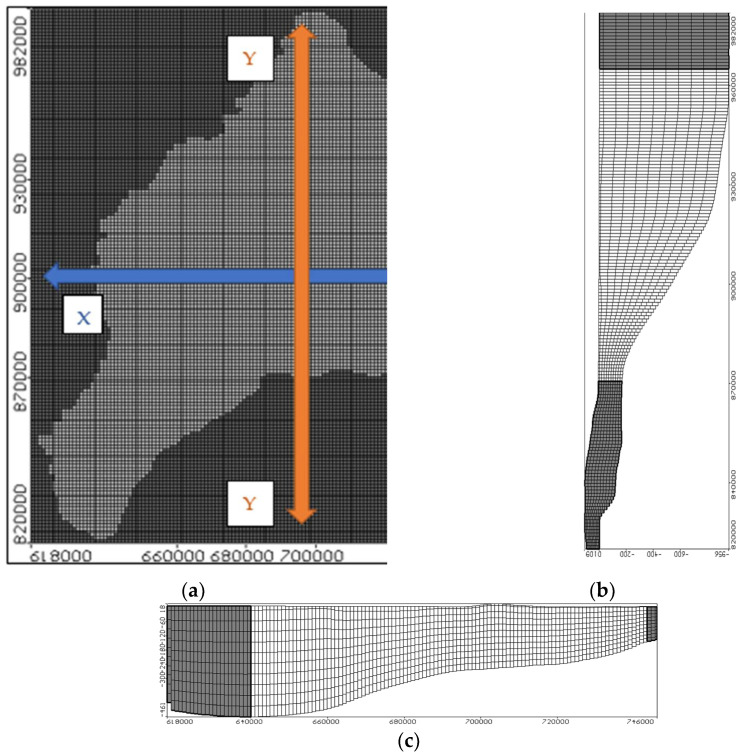
Model geometry of the ENDA: (**a**) model domain (**b**) cross-section in the X-direction and (**c**) cross section in the Y-direction.

**Figure 4 ijerph-19-14929-f004:**
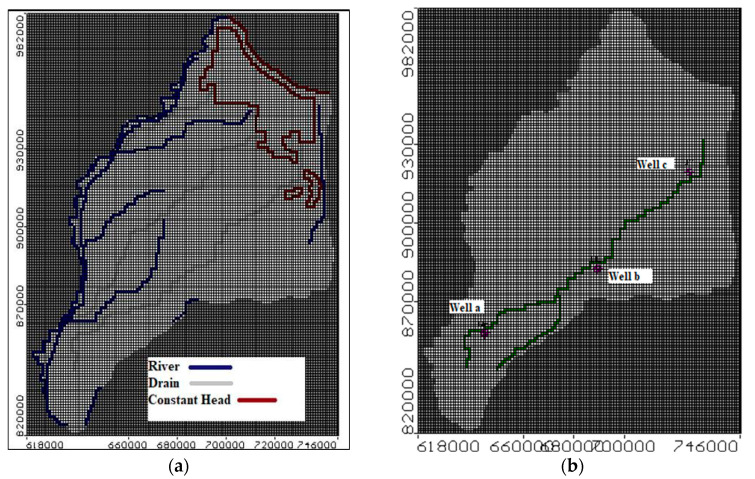
Boundary conditions of the ENDA: (**a**) head boundary conditions, (**b**) contaminant concentration boundary conditions and location of observation wells to evaluate COD concentration.

**Figure 5 ijerph-19-14929-f005:**
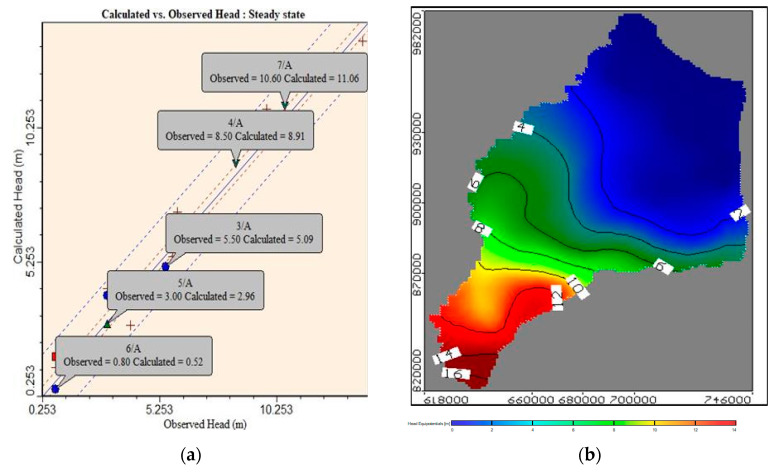
Results of model calibration: (**a**) calculated and observed head and (**b**) areal section of groundwater flow head in the ENDA.

**Figure 6 ijerph-19-14929-f006:**
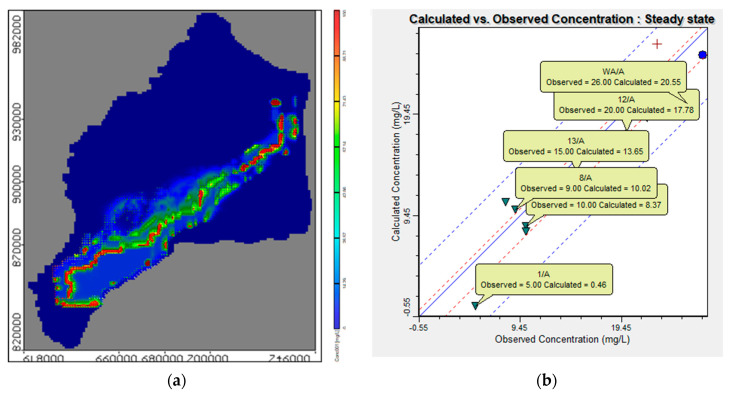
(**a**) Aerial view of COD distribution in the ENDA. (**b**) The difference between calculated and observed concentration.

**Figure 7 ijerph-19-14929-f007:**
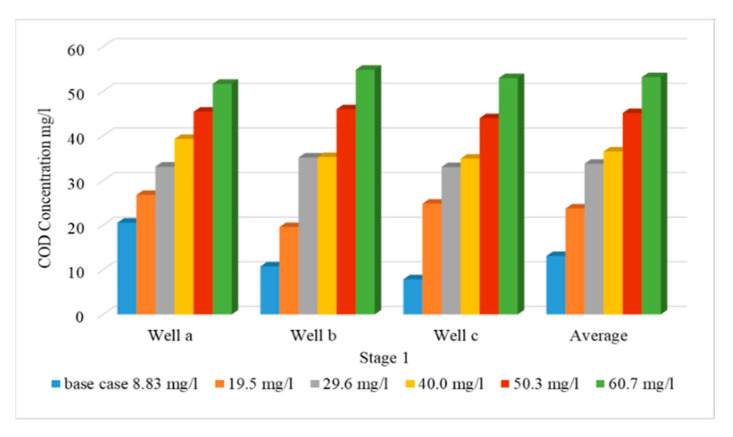
Comparison between COD concentration at observation wells for different scenarios (2–6) of stage 1.

**Figure 8 ijerph-19-14929-f008:**
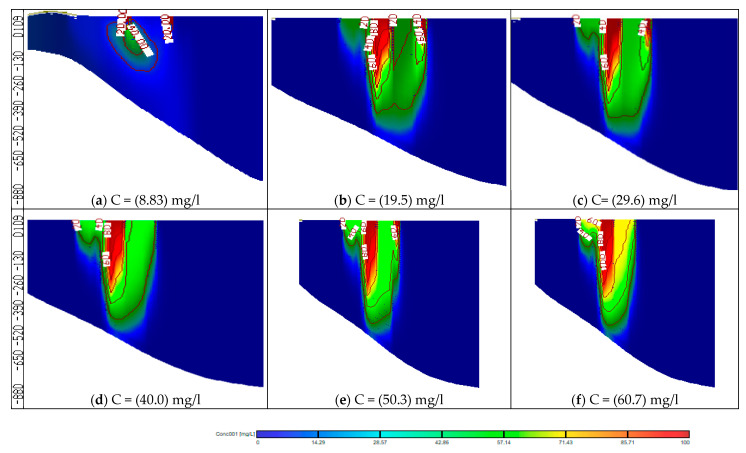
Distribution of COD in the ENDA for different scenarios (2−6) of stage 1.

**Figure 9 ijerph-19-14929-f009:**
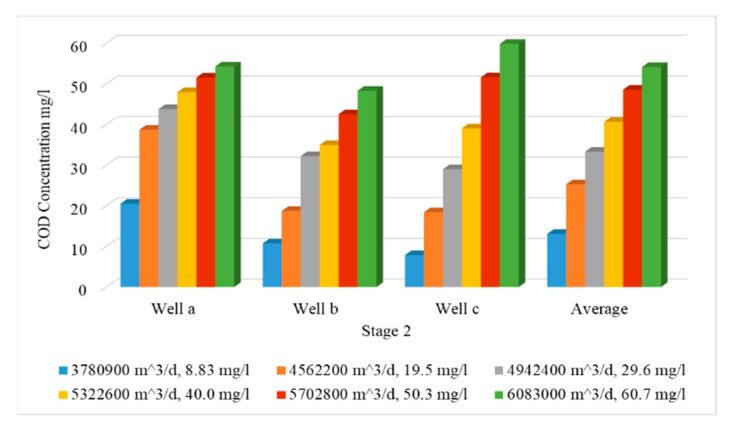
Comparison between COD concentration at observation wells for different scenarios (7–11) of stage 2.

**Figure 10 ijerph-19-14929-f010:**
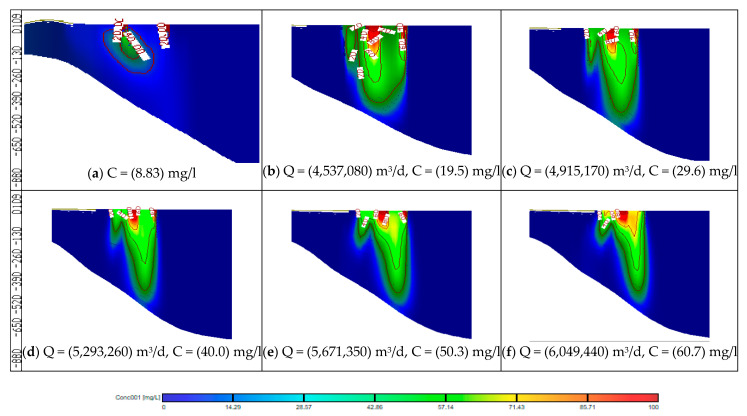
Distribution of COD in the ENDA for different scenarios (7–11) of stage 2.

**Figure 11 ijerph-19-14929-f011:**
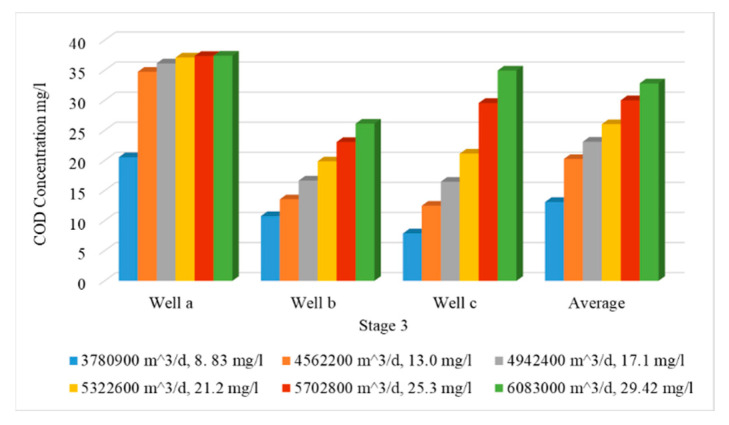
Comparison between COD concentration at observation wells for different scenarios (12–16) of stage 3.

**Figure 12 ijerph-19-14929-f012:**
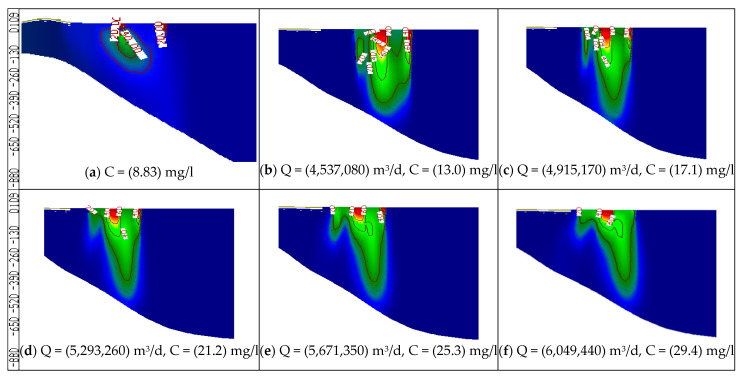
Distribution of COD in the ENDA for different scenarios (12−16) of stage 3.

**Figure 13 ijerph-19-14929-f013:**
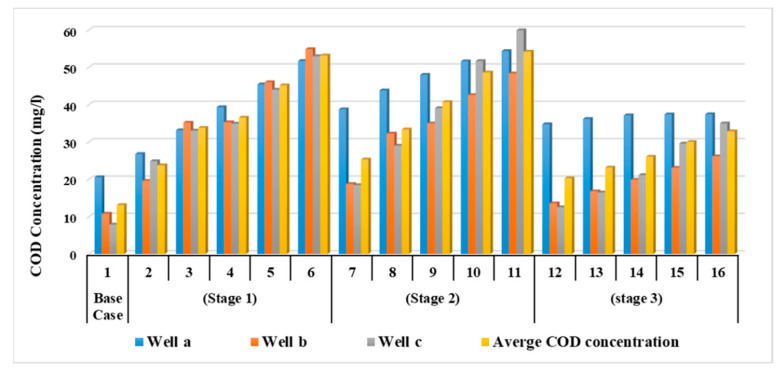
Comparison between COD concentration at observation wells for different scenarios of the three stages (2–16) with the base case (1).

**Table 1 ijerph-19-14929-t001:** Hydraulic parameters of the ENDA.

Layer No.		1	2 to 5	6 to 9	10, 11
Layer type		Clay	Fine sand with lenses of clay	Course sand quaternary	Graded sand and gravel
Hydraulic conductivity (K) m/day	(K_h_) horizontal	0.10–0.25	5–20	20–75	75–100
	(K_v_) vertical	0.01–0.025	0.5–2	2–7.5	7.5–10
Specific storage (S_S_) (1/m)		0.001	0.005	0.0025	0.0005
Specific yield (S_y_)		0.10	0.15	0.18	0.20
Effective porosity (n_eff_) %		50–60	30	25	20

**Table 2 ijerph-19-14929-t002:** The Total Inflow and Outflow Model.

**Parameters**	Inflow	Outflow	Difference
**Constant head**	239,290	−886,710	−647,420
**Wells**	0	−3,780,900	−3,780,900
**Drains**	0	−68,618	-68,618
**Recharge**	4,299,700	0	4,299,700
**River leakage**	340,530	−143,360	197,170
**Total (m^3^/d)**	4,879,520	−4,879,588	−68

**Table 3 ijerph-19-14929-t003:** Proposed scenarios applied for the ENDA.

	**-**	**Scenario No.**	**Q_pump_ (m^3^/d)**	**R (m^3^/d)**		**C (mg/L)**
	Base Case	1	3,780,900	4,376,900		8.83
Investigation	Untreated wastewater	2	3,780,900	Freshwater	Untreated wastewater	19.5
				90%	10%	
		3		80%	20%	29.6
		4		70%	30%	40.0
		5		60%	40%	50.3
		6		50%	50%	60.7
	Untreated wastewater with over pumping	7	4,537,080 (20%)	90%	10%	19.5
		8	4,915,170 (30%)	80%	20%	29.6
		9	5,293,260 (40%)	70%	30%	40.0
		10	5,671,350 (50%)	60%	40%	50.3
		11	6,049,440 (60%)	50%	50%	60.7
Management	Treated wastewater with over pumping	12	4,537,080 (20%)	Freshwater	Treated wastewater	13.0
				90%	10%	
		13	4,915,170 (30%)	80%	20%	17.1
		14	5,293,260 (40%)	70%	30%	21.2
		15	5,671,350 (50%)	60%	40%	25.3
		16	6,049,440 (60%)	50%	50%	29.4

**Table 4 ijerph-19-14929-t004:** COD concentration for different scenario.

Stage	Scenario No	Well-a (mg/L)	Well-b (mg/L)	Well-c (mg/L)	Average COD Concentration (mg/L)
Base Case	1	20.55	10.76	7.88	13.10
Stage 1 (Untreated wastewater)	2	26.78	19.57	24.83	23.73
	3	33.11	35.15	33.02	33.76
	4	39.31	35.26	34.90	36.49
	5	45.41	45.99	43.98	45.13
	6	51.66	54.82	52.96	53.13
Stage 2 (Untreated wastewater with over-pumping)	7	38.73	18.72	18.43	25.30
	8	43.79	32.24	28.99	33.34
	9	47.96	34.97	39.05	40.66
	10	51.57	42.55	51.64	48.60
	11	54.32	48.32	59.87	54.17
Stage 3 (Treated wastewater with over-pumping)	12	34.74	13.55	12.49	20.26
	13	36.15	16.76	16.47	23.13
	14	37.11	19.86	21.13	26.03
	15	37.37	23.07	29.56	30.00
	16	37.41	26.13	34.96	32.83

## Data Availability

Not applicable.

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
