# Peer review of "Groundwater Quality Modeling and Mitigation from Wastewater Used in Irrigation, a Case Study of the Nile Delta Aquifer in Egypt"

_ijerph, 2022, doi:10.3390/ijerph192214929_

Round 1

Reviewer 1 Report

Manuscript (ID: ijerph-1989545)_

Title: Groundwater quality investigation in the Eastern Nile Delta Aquifer of Egypt due to using polluted drains water for irrigation

General Comments:

In this study, the author use visual MODFLOW model to simulate groundwater flow and contaminant transport. Groundwater quality variation at three different stages was characterized based on COD concentrations.

However, I cannot found out the novelty and something new or different from other studies relevant to the research topic. Besides, inappropriate language, unprofessional/odd expressions and grammatical mistakes can be easily found in the full text.

The present version is far from being ready for submission. Many supplements and modifications are required to improve the article quality. Just as important as the research contents, the language and grammar should be checked and revised carefully. Therefore, I recommend Major Revision. The author should revise the manuscript carefully according to specific comments below.

Specific Comments:

1.      Abstract- “ppm” should be “mg/L”. The author should use SI unit in scientific articles.

2.      Introduction section should be more refined. The author should focus on the status of your research topic. Additionally, advantages and disadvantages of wastewater irrigation should also be compared, especially in your study area/country.

3.      L75, m3 should be m3.

4.      L 159, Km2 should be km2. The author should check superscript/ subscript carefully in the full manuscript!

5.      2.1. Study Area- Lack of necessary hydrogeological condition, e.g. buried condition and water abundance of aquifer, groundwater recharge-runoff-discharge condition, key hydrogeological parameters, groundwater recharge capacity, groundwater resource and groundwater yield.

6.      Figure 3- necessary monitoring wells for groundwater level and groundwater quality should be marked in this figure.

7.      2.2.6. Model calibration- 1) supplementary discussion on fitting result of groundwater equilibrium is required. 2) Lack of necessary fitting figure of measured groundwater level in typical monitoring wells and simulated groundwater level. 3) Add calibration figure of measured groundwater COD and simulated groundwater COD in MT3D model.

8.      Figure 6b, lack of measured groundwater flow field.

9.      Report all the COD concentrations to three significant figures or one decimal digit.

10.   Table 2- 1) improper hypothesis of invariable recharge amount. When groundwater yield increased, it could cause extra lateral recharge to groundwater (recharge increment). 2) Provide aquifer boundary conditions of the study area. 3) Non-correspondence in R (mm/yr) column and Scenario No column; delete Total column. 4) Do groundwater exploitation and wastewater irrigation under different scenario cover the whole study area uniformly? If so, does it meet the actual hydrogeological conditions and present irrigation status?

11.   Figure 10- “ppm” should be “mg/L”, m3 should be m3.

12.   Figure 11- it is hard to read.

13.   Table 3- supplementary analysis of difference between COD in scenario 2~16 and that in scenario 1 is required. Besides, discussion on difference of DOC concentrations in Well a, b and c is also required.

14.   References- the author should check the reference format very carefully. non-uniform description format of academic dissertation and journal title (full title or abbreviation) can be easily found in the reference list. Use uniform reference format.

15. All the figures- The author should upload a manuscript that contains higher quality of figures. use unified fonts and font size. The current version is hard to read.

Author Response

Reviewer #1:

In this study, the author use visual MODFLOW model to simulate groundwater flow and contaminant transport. Groundwater quality variation at three different stages was characterized based on COD concentrations. However, I cannot found out the novelty and something new or different from other studies relevant to the research topic. Besides, inappropriate language, unprofessional/odd expressions and grammatical mistakes can be easily found in the full text.

Thanks, the novelty of the paper has been highlighted in the  abstract, end of introduction and conclusion. The language has been checked, revised, and corrected in the full text.

The present version is far from being ready for submission. Many supplements and modifications are required to improve the article quality. Just as important as the research contents, the language and grammar should be checked and revised carefully. Therefore, I recommend Major Revision. The author should revise the manuscript carefully according to specific comments below.

Thanks, the article quality has been improved based on comments and recommendations from three reviewers and the language and grammar have been checked and revised carefully.

Specific comments.

Comment 1: Abstract- “ppm” should be “mg/L”. The author should use SI unit in scientific articles.

Response: Has been changed in all the text.

Comment 2: Introduction section should be more refined. The author should focus on the status of your research topic. Additionally, advantages and disadvantages of wastewater irrigation should also be compared, especially in your study area/country.

Response: The introduction section is refined, and new references have been added. Also, advantages and disadvantages of wastewater irrigation have been added  

Comment 3: L75, m3 should be m3.

Response: Has been changed in all the text.

Comment 4: L 159, Km2 should be km2. The author should check superscript/ subscript carefully in the full manuscript!

Response: Has been changed and superscript/subscript have been checked in all the text.

Comment 5: 2.1. Study Area- Lack of necessary hydrogeological condition, e.g. buried condition and water abundance of aquifer, groundwater recharge-runoff-discharge condition, key hydrogeological parameters, groundwater recharge capacity, groundwater resource and groundwater yield.

Response: The necessary hydrogeological conditions have been added to the text and shown in Figure 1.b.

Comment 6: Figure 3- necessary monitoring wells for groundwater level and groundwater quality should be marked in this figure.

Response: The observation wells for contamination and head are added in the study area in Figure A3 (appendix).

Comment 7: 2.2.6. Model calibration- 1) supplementary discussion on fitting result of groundwater equilibrium is required. 2) Lack of necessary fitting figure of measured groundwater level in typical monitoring wells and simulated groundwater level. 3) Add calibration figure of measured groundwater COD and simulated groundwater COD in MT3D model.

Response:

  • Results of groundwater equilibrium are shown in Table 2.
  • Measured groundwater level is shown in Figure 5.
  • Added in Figure 6.

Comment 8: Figure 6b, lack of measured groundwater flow field.

Response: Has been added modified and explained in the text.

Comment 9: Report all the COD concentrations to three significant figures or one decimal digit.

Response: Thanks, done and corrected.

Comment 10: Table 2- 1) improper hypothesis of invariable recharge amount. When groundwater yield increased, it could cause extra lateral recharge to groundwater (recharge increment). 2) Provide aquifer boundary conditions of the study area. 3) Non-correspondence in R (mm/yr) column and Scenario No column; delete Total column. 4) Do groundwater exploitation and wastewater irrigation under different scenario cover the whole study area uniformly? If so, does it meet the actual hydrogeological conditions and present irrigation status?

Response:

  • Yes, this is considered in the model for the water budget by the constant head, please see Table 2.2
  • Boundary conditions in the model to develop the proposed scenarios were remained constant while the changed boundary conditions are presented in Table 3.
  • Thanks, done and corrected.
  • Yes, as presented in Figure 2.

Comment 11: Figure 10- “ppm” should be “mg/L”, m3 should be m3.

Response: Has been changed in all the text.

Comment 12: Figure 11- it is hard to read.

Response: The figure has been corrected.

Comment 13: Table 3- supplementary analysis of difference between COD in scenario 2~16 and that in scenario 1 is required. Besides, discussion on difference of COD concentrations in Well a, b and c is also required.

Response: The necessary supplementary analysis has been added to the results and discussion.

Comment 14: References- the author should check the reference format very carefully. non-uniform description format of academic dissertation and journal title (full title or abbreviation) can be easily found in the reference list. Use uniform reference format.

Response: The references format has been checked and reviewed.

Comment 15: All the figures- The author should upload a manuscript that contains higher quality of figures. use unified fonts and font size. The current version is hard to read.

Response: Thanks, the figures have been modified and higher quality figures have been added.

Reviewer 2 Report

This paper modelled the impact of using polluted drains water for irrigation on groundwater quality in the Eastern Nile Delta Aquifer. However, this is a long manuscript in which the objectives are not clear and the authors have not arrived at some usable conclusions. I think that the paper is not acceptable for publication. The general comments are as follows:

1.        The paper is not investigation, just modeling the groundwater quality. The title can be changed to ‘Groundwater quality modelling ……

2.        The objectives of this study need to be better given. What is the main reason for carrying out this research? Why did you select three stages?

3.        Discussion needs to be improved to explain the relevancy of these findings. Also, results need to be related with research carried out in the Nile Delta Aquifer and elsewhere. This is rather short in the current version.

Author Response

Reviewer #2: 

Groundwater is an essential supply of freshwater. But the usage of wastewater can potentially pose environmental problems by contaminating soil and groundwater. This study investigated the impact of using polluted drains water for irrigation on groundwater quality in the Eastern Nile Delta Aquifer. However, water quality is hardly reflected just by one indicator (COD). Additionally, the descriptions on sections 1 and 2 are too long and tedious to highlight essential points. On the contrary, section 4, which should be the most important part, is too shallow. I think this manuscript needs a major revision with some reworking and deep discussions before considering publication in this journal.

Thanks, the paper has been revised according to comments and recommendations from three reviewers and all the comments have been addressed which improved the paper quality.

Some suggestions for authors:

Comment 1: The title needs to be improved to better summarize the research.

Response: The title has been changed to “Groundwater quality modelling and mitigation from wastewater used in irrigation, a case study of the Nile Delta aquifer in Egypt”

Comment 2: The abstract should more describe significant new results of this study and their implications.

Response: The abstract has been changed and new part has been added to describe significant new results of this study and their implications.

 Comment 3: The introduction should clearly and concisely explain the motivation and significances of this study and discuss the relationships of this study with previously published work, instead of simply reiterating or providing a complete literature survey (e.g., the last several paragraphs). The first eight paragraphs are verbose and don't matter too much with contaminated groundwater.

Response: The introduction has been modified and more discussion has been added. The relationships of this study with previously published work have been discussed in more details and more recent references have been added.

Comment 4:  In Figure 1, what the means of various color?

Response: The figure has been changed. 

Comment 5: Many contents and figures in section 2 should be moved to supporting materials to make the texts brevity.

Response: Thanks, an appendix has been added and some figures have been moved to the appendix to make the texts brevity.

Comment 6: Some important conclusive statements, in both the introduction and discussion, lack reliable references to support.

Response: The introduction and discussion have been modified and recent references have been added.

Comment 7: Many sentences in the discussion are repeated or should belong to the results.

Response: The discussion and the result are reviewed and modified to avoid any repeating  words.

Comment 8: Line 340: What are the selection criteria for the 3 observation wells? Why the other boundary has not be selected in Figure 9?

Response: The selection criteria for the 3 observation wells has been clarified in (line 448-451). “ Three observation wells a, b and c were distributed in the study area. The first well (a) was located in the lower part, the second well (b) was located in the middle part  and the third well (c) was located in the upper part”.

Comment 9: The conclusion is not summarized properly. The impact of over-pumping should be added to the conclusion.

Response: The conclusion has been summarized and impact of over-pumping has been clarified in the discussion and conclusion.

Comment 10:  This manuscript suffers from serious “use of English” problems, e.g., there is a full stop missing at the end of the sentence in line 154; a space symbol is missing between ‘14’ and ‘presents’.

Response: Thanks, the manuscript has been revised and all English errors have been corrected.

Reviewer 3 Report

Groundwater is an essential supply of freshwater. But the usage of wastewater can potentially pose environmental problems by contaminating soil and groundwater. This study investigated the impact of using polluted drains water for irrigation on groundwater quality in the Eastern Nile Delta Aquifer. However, water quality is hardly reflected just by one indicator (COD). Additionally, the descriptions on sections 1 and 2 are too long and tedious to highlight essential points. On the contrary, section 4, which should be the most important part, is too shallow. I think this manuscript needs a major revision with some reworking and deep discussions before considering publication in this journal.

Some suggestions for authors:

1.       The title needs to be improved to better summarize the research.

2.       The abstract should more describe significant new results of this study and their implications.

3.       The introduction should clearly and concisely explain the motivation and significances of this study and discuss the relationships of this study with previously published work, instead of simply reiterating or providing a complete literature survey (e.g., the last several paragraphs). The first eight paragraphs are verbose and don't matter too much with contaminated groundwater.

4.       In Figure 1, what the means of various color?

5.       Many contents and figures in section 2 should be moved to supporting materials to make the texts brevity.

6.       Some important conclusive statements, in both the introduction and discussion, lack reliable references to support.

7.       Many sentences in the discussion are repeated or should belong to the results.

8.       Line 340: What are the selection criteria for the 3 observation wells? Why the other boundary has not be selected in Figure 9?

9.       The conclusion is not summarized properly. The impact of over-pumping should be added to the conclusion.

10.    This manuscript suffers from serious “use of English” problems, e.g., there is a full stop missing at the end of the sentence in line 154; a space symbol is missing between ‘14’ and ‘presents’.

Author Response

Reviewer #3:

This paper modelled the impact of using polluted drains water for irrigation on groundwater quality in the Eastern Nile Delta Aquifer. However, this is a long manuscript in which the objectives are not clear and the authors have not arrived at some usable conclusions. I think that the paper is not acceptable for publication. The general comments are as follows:

Thanks, the paper quality has been improved based on the comments and recommendations from 3 reviewers by adding aims, and objectives in abstract (lines 19-24). Also, the results and discussion parts have been modified and main findings of the study were clarified in the abstract and conclusion.

Comment 1: The paper is not investigation, just modeling the groundwater quality. The title can be changed to ‘Groundwater quality modelling ……’

Response: Thanks, the title has been changed to “Groundwater quality modelling and mitigation from wastewater used in irrigation, a case study of the Nile Delta aquifer in Egypt”. This was also recommended by other reviewers, thanks a lot for this valuable comment.

Comment 2: The objectives of this study need to be better given. What is the main reason for carrying out this research? Why did you select three stages?

Response: The main aims and objectives of this research has been clarified in the abstract, end of introduction and conclusion. The study was carried out on three stages because each stage includes assessing different impacts on groundwater quality. The first stage investigated the impact of using untreated wastewater for irrigation due to shortage of fresh water. The second stage observed the influence of over-pumping with using untreated wastewater in irrigation due to population increase. The third stage included the mitigation of the previous two stages to improve the impact on groundwater quality through mixing treated wastewater with freshwater to improve the freshwater quantity.

Comment 3: Discussion needs to be improved to explain the relevancy of these findings. Also, results need to be related with research carried out in the Nile Delta Aquifer and elsewhere. This is rather short in the current version.

Response: Thanks, the discussion has been improved and necessary related research have been added.

Round 2

Reviewer 1 Report

(1)Figure A3 needs to add the observation well number, corresponding to Figure 4.

(2)In Figure A1, the legends should be arranged in order of the amount of recharge. What is the scale on the far right of the abscissa?

(3)In 2.2.6. Model calibration, dynamic curves of water level and COD measured and simulated values for a typical observation well during the simulation period need to be supplemented.

Author Response

Reviewer #1:

Comment 1: Figure A3 needs to add the observation well number, corresponding to Figure 4.

Response: Thanks, Figure A3 has been modified and observation well number have been included.

Comment 2: In Figure A1, the legends should be arranged in order of the amount of recharge. What is the scale on the far right of the abscissa?

Response: Thanks, the legend in Figure A1 has been arranged from smallest to largest values of recharge. The legend on the right describes the value of recharge rate and the color of each value.

Comment 3: In 2.2.6. Model calibration, dynamic curves of water level and COD measured and simulated values for a typical observation well during the simulation period need to be supplemented.

Response: This has been shown in Figure 5.a and Figure 6.b, both at steady state conditions as explained in the text (lines 286-339) 

Reviewer 2 Report

The language and grammar should be revised from Language Polishing Service. 

Author Response

Reviewer #2: 

Comment: The language and grammar should be revised from Language Polishing Service. Response: The language and grammar have been checked, revised, and corrected in the full text.

We would like to thank the reviewer for their valuable comments that have certainly improved the quality of the paper.
